# Intervention Target Estimation in the Presence of Latent Variables

**Burak Varıcı**[1]   **Karthikeyan Shanmugam** [*2]   **Prasanna Sattigeri**[2]   **Ali Tajer**[1]

[1]Rensselaer Polytechnic Institute
[2]IBM Research AI

## Abstract

This paper considers the problem of estimating unknown intervention targets in causal directed acyclic graphs from observational and interventional data in the presence of latent variables. The focus is on linear structural equation models with soft interventions. The existing approaches to this problem involve performing extensive conditional independence tests, and they estimate the unknown intervention targets alongside learning the structure of the causal model in its entirety. This joint learning approach results in algorithms that are not scalable as graph sizes grow. This paper proposes an approach that does not necessitate learning the entire causal model and focuses on learning only the intervention targets. The key idea of this approach is leveraging the property that interventions impose sparse changes in the precision matrix of a linear model. The proposed framework consists of a sequence of precision difference estimation steps. Furthermore, the necessary knowledge to refine an observational Markov equivalence class (MEC) to an interventional MEC is inferred. Simulation results are provided to illustrate the scalability of the proposed algorithm and compare it with those of the existing approaches.

## 1 INTRODUCTION

Enabling modern machine learning systems to reason involves predicting the effect of an intervention and counterfactual estimation [Pearl, 2009]. Forming such predictions crucially depends on the knowledge of causal models [Pearl and Mackenzie, 2018]. One approach to represent causal knowledge is through a causal Bayesian network, which is a directed graphical model specified by a directed acyclic graph (DAG). The nodes of a DAG represent random variables, and its directed edges represent the cause-and-effect relationships among the random variables. Such a model facilitates factorizing the observed distribution, where each factor is a conditional distribution of a variable given its causal parents. These conditionals specify the local causal mechanisms of the variables. However, based on purely observational data, a causal DAG is identifiable only up to an equivalence class of DAGs. Such uncertainty is because different DAGs can encode different ways of factorizing the same observed distribution into conditionals. The equivalence class of DAGs that can be identified from the observational data alone is called the Markov equivalence class (MEC) [Peters et al., 2017].

To reduce the ambiguity in the MEC obtained from the observational data, interventional data can be leveraged. Intervening on a variable refers to modifying the causal mechanism (the conditional distribution) that connects this variable and its parents in the true causal DAG while leaving the other factors unchanged. The combination of observational and interventional data reduces the number of possible factorizations that are consistent with both data types. In this paper, we perform *soft* interventions. A soft intervention induces a change in the causal mechanism by replacing it with a different one without requiring the causal effects on the target node to be removed. While hard interventions, e.g., assigning fixed values to intervention targets, can be performed too, there are applications in which soft interventions are better suited for modeling the experiments. For instance, soft interventions can effectively model altering the gene expressions for cellular reprogramming [Zhang et al., 2021].

In a broad range of applications, when interventional data is available, the variables whose causal mechanisms have been changed, called the *intervention targets*, are unknown. For instance, there is a recent growing interest in using causal discovery for fault localization in microservices systems in cloud-native applications [Bogatinovski et al., 2021,

---

*Author is currently affiliated with Google Research India. The work was done while the author was at IBM Research AI, NY.

*Accepted for the 38th Conference on Uncertainty in Artificial Intelligence* (UAI 2022).

Aggarwal et al., 2020]. These systems are built as an interconnected set of loosely coupled services across various layers [Kim et al., 2013, Mariani et al., 2018]. Such systems are vulnerable to unwanted changes (e.g., equipment failure and attacks). During the faulty operation of these systems, it is imperative to localize the faults quickly. The root causes of the faulty operations are modeled as interventions to the system. Hence, the data is collected under (unknown) faults, rendering fault localization a causal discovery task from interventional data of unknown intervention targets. Furthermore, a fault in the operation of a node, e.g., a delay, is closer to soft interventions than to hard interventions since the causal parents can still affect the operation of the node. Another example is gene knockout experiments in biology. In these experiments, a target set of genes is knocked out in an assay, and gene expressions are collected. These are known to affect off-target genome sites [Fu et al., 2013]. Sometimes drugs are injected into protein signaling networks, and the expression levels are measured. In these settings, the intervention targets are unknown [Sachs et al., 2005, Ness et al., 2017].

Identifying unknown intervention targets in fully observed graphs was recently explored [Varici et al., 2021]. However, in this study, all variables of a true causal DAG are typically not observed. This induces confounding between observed variables due to unobserved or latent variables. A model with such confounding is called a *causally insufficient* model. Recent studies have characterized the interventional MEC for causally insufficient models and have provided algorithms for learning their structures. These algorithms leverage invariance testing and conditional independence testing by using both interventional and observational data and accommodate both settings of known and unknown intervention targets [Mooij et al., 2020, Kocaoglu et al., 2019, Jaber et al., 2020]. In these algorithms, the intervention targets are usually learned along with the interventional MEC. In this paper, we focus on the following question: **is there an efficient way to learn only the intervention targets given interventional and observational data?**

**Our Contributions:** We address the above question in linear structural equation models (SEMs) under soft interventions. We first show that the difference in the precision matrices of the interventional datasets can be used to deduce the intervention status of a node. Next, we use the fact that these precision differences have sparse support to narrow down our interest to the nodes directly affected by the interventions. Then, we show how to refine this sparse set by repeated precision difference estimations to obtain the intervention targets. In the process, we also infer the causal knowledge newly induced by the interventions. Finally, using these elements, we propose a scalable algorithm to estimate the intervention targets.

There are two studies whose scopes are close to that of this paper. Jaber et al. [2020] characterize the interventional MEC for soft interventions and proves that the intervention targets can be identified only up to a superset that they graphically describe. Noting this result, in this paper, we focus on estimating this superset, which we call the *effective intervention targets*. In a different study, Varici et al. [2021] address a related problem. However, their method is limited to only causally sufficient models. We present theoretical results for causally insufficient models, which are non-trivial generalizations that combine the precision difference approach to the problem and the graphical characterization of the soft interventions on causally insufficient models.

The existing interventional causal discovery algorithms for insufficient systems jointly learn the causal structure and the intervention targets. These approaches require performing a significant number of conditional independence and invariance tests, a major impediment to these algorithms for being scalable to large graphs [Jaber et al., 2020]. However, unlike interventional causal discovery, there exist highly efficient algorithms for causal discovery with observational data. One of the byproducts of our results is that our scalable algorithm for intervention target discovery can be used in conjunction with any observational learning algorithm for insufficient systems to refine the observational MEC efficiently to an interventional MEC. Finally, we perform experiments on real and synthetic datasets to illustrate the scalability of the proposed algorithm.

## 2 RELATED WORK

At its core, this paper infers causal knowledge from interventional settings through an invariance criterion. The existing literature on related topics is discussed next.

**Interventional causal learning for causally sufficient systems.** There is extensive literature on interventional learning for causally sufficient models. Among them, Eaton and Murphy [2007] proposed a dynamic programming approach to interventional learning. Hauser and Bühlmann [2012] considers the interventional MEC under hard interventions and provides a score-based algorithm for interventional learning. Rothenhäusler et al. [2015] learn causal cyclic graphs using shift interventions. Ghassami et al. [2018] consider multi-domain data without explicitly formulating the different domains via interventions. Its method estimates the causal order by generalizing the invariance of parameters to the independence of the changes in the parameters across domains. Huang et al. [2020] use the distribution shifts that can be the results of interventions to determine the causal directions. Their method works under a pseudo-causal sufficiency condition in which the values of the unobserved confounders are fixed in each domain. Yang et al. [2018] characterize interventional MEC under hard and soft interventions using invariance testing and provides a learning

algorithm when the intervention targets are known. The algorithm of Squires et al. [2020] greedily searches over the space of permutations to score DAGs when the intervention targets are unknown. Ke et al. [2019] and Brouillard et al. [2020] leverage differentiable methods through continuous optimization to learn the causal structure from interventional data. For linear SEMs and causally sufficient models, Wang et al. [2018] propose to learn the difference graph, which is the set of edge weights in the linear SEM that have been changed across two environments. Ghoshal et al. [2021] leverage precision difference estimates to address the same problem under more stringent assumptions. Varici et al. [2021] use precision difference estimates and achieves a higher level of scalability through a hierarchical grouping of the nodes.

**Learning from observational data for causally insufficient systems.** The fast causal inference (FCI) algorithm of Spirtes et al. [2000] is a classic constraint-based method for learning causally insufficient models from observational data. Many efficient variants such as the really fast causal inference (RFCI) algorithm of Colombo et al. [2012] and the greedy fast causal inference (GFCI) algorithm of Ogarrio et al. [2016] have been proposed to improve scalability. Bernstein et al. [2020] extend the greedy permutation search to partially ordered sets to include the effects of latent variables in ordering.

**Learning from interventions on causally insufficient systems.** Triantafillou and Tsamardinos [2015] consider multiple interventions for causally insufficient systems. Their algorithm applies ideal hard interventions and provides a solution based on constraint satisfaction and conditional independence testing. Mooij et al. [2020] propose a joint causal inference framework to pool interventional datasets to learn the causal graph. Jaber et al. [2020] characterize the interventional MEC and propose a variant of FCI to learn from soft interventional data in causally insufficient systems. The key shortcoming of these methods is that their runtime becomes prohibitive for large graphs.

## 3 PRELIMINARIES

We introduce some concepts and notations pertinent to causal discovery in causally insufficient systems.

Let $\mathcal{D} \triangleq (\mathbf{W}, \mathbf{E})$ denote a causal graph in which $\mathbf{W}$ represents the set of nodes and $\mathbf{E}$ represents the set of edges. Denote the number of nodes by $p \triangleq |\mathbf{W}|$. We associate the random variable $X_i$ to node $i$, for $i \in [p] \triangleq \{1, \ldots, p\}$, and accordingly define the random vector $X \triangleq (X_1, \ldots, X_p)^{\top 1}$. We consider a linear SEM, according to which

$$X = B^\top X + \epsilon , \tag{1}$$

---
[1]Throughout the paper, we use $X_i$ to represent node $i \in [p]$.

where $B \in \mathbb{R}^{p \times p}$ is the edge weights matrix in which $B_{i,j} \neq 0$ if and only if $X_i \rightarrow X_j \in \mathbf{E}$. The random noise vector $\epsilon \in \mathbb{R}^{p \times 1}$ has zero mean with covariance matrix $\Omega \triangleq \mathsf{diag}(\sigma_1^2, \ldots, \sigma_p^2)$. We denote the covariance matrix of $X$ by $\Sigma$, and the precision matrix by $\Theta$, which satisfies $\Theta = (I - B)\Omega^{-1}(I - B)^\top$. For the entries of $\Theta$ we have

$$\Theta_{i,j} = -\frac{B_{i,j}}{\sigma_j^2} - \frac{B_{j,i}}{\sigma_i^2} + \sum_{k \in \mathsf{ch}(i) \cap \mathsf{ch}(j)} \frac{B_{i,k} B_{j,k}}{\sigma_k^2} , \quad \forall i \neq j , \tag{2}$$

$$\Theta_{i,i} = \frac{1}{\sigma_i^2} + \sum_{j \in \mathsf{ch}(i)} \sigma_j^{-2} B_{i,j}^2 , \qquad \forall i \in [p] , \tag{3}$$

where $\mathsf{ch}(i)$ denotes the set of children of node $i \in [p]$. In the causal graph $\mathcal{D}$, we have two sets of nodes: a set of observed variables denoted by $\mathbf{V}$, and a set of latent variables denoted by $\mathbf{L}$. Clearly, $\mathbf{V} \cup \mathbf{L} = \mathbf{W}$. The observational data, consequently, is represented by $\{X_i : i \in \mathbf{V}\}$.

From the observational data alone, a DAG with only observed variables can be identified up to its MEC [Verma and Pearl, 1992]. For causally insufficient systems with latent variables $\mathbf{L}$, we can only describe the MEC in terms of a family of graphs called *maximal ancestral graphs* (MAGs), which we formally specify later in this section. The MAG associated with $\mathbf{V}$ represents the pairwise ancestral and confounding relationships among the observed variables $\{X_i : i \in \mathbf{V}\}$ that cannot be made conditionally independent. Therefore, for the true causal graph $\mathcal{D}$, there exists a unique MAG. This MAG cannot be identified uniquely. However, it is possible to recover it up to a family of equivalent MAGs that contains the true one. Next, we describe how a MAG is obtained from a DAG and then proceed to describe the MEC of MAGs and how they are represented.

**Mixed Graphs:** From a structure learning perspective, causally insufficient systems are often represented by *mixed* graphs. A mixed graph can contain both directed ($\rightarrow$) and bi-directed ($\leftrightarrow$) edges. In our notations, we use $\leftarrow\circ$ to emphasize that an edge represents either a directed or a bi-directed edge. If there is a directed path from node $A$ to node $B$, then $A$ is an ancestor of $B$, and $B$ is a descendant of $A$. Bi-directed edges create *spouses*, that is, $A$ is a spouse of $B$ if $A \leftrightarrow B$ is present. A node on a path is a *collider* if both of its edges on the path are into the node. A triple $\langle X, Y, Z \rangle$ is an *unshielded collider* if $X \circ\rightarrow Y \leftarrow\circ Z$, and $X$ and $Z$ are not adjacent. A path $\langle X, \ldots, W, Z, Y \rangle$ is a *discriminating path* for $Z$ if every node between $X$ and $Z$ is a collider on the path, and is also a parent of $Y$. An *inducing path* relative to $\mathbf{L}$ is a path on $\mathcal{D}$ such that on this path, every non-endpoint node $X \notin \mathbf{L}$ is a collider on the path, and every collider is an ancestor of an endpoint of the path.

**Maximal Ancestral Graphs:** Consider the causal graph $\mathcal{D} = (\mathbf{V} \cup \mathbf{L}, \mathbf{E})$. A unique mixed graph called the MAG [Richardson and Spirtes, 2002] $\mathcal{M}_\mathcal{D}$ over $\mathbf{V}$ has the following three properties: (i) in a MAG, there exists an edge between two nodes if and only if their associated variables

cannot be made conditionally independent (or d-separated) by conditioning on any subset of observed variables in the true $\mathcal{D}$; (ii) if there is an edge in the skeleton that represents the ancestral relationships among the variables in $\mathbf{V}$ in the true $\mathcal{D}$ [Zhang, 2008], then a directed edge is used to represent this edge; and (iii) if there is an edge in a MAG that connects two variables that do not have any ancestral relationship in $\mathcal{D}$, then a bi-directed edge $\leftrightarrow$ is used to represent it. We note that the relationships between DAGs and MAGs are many-to-one, i.e., different DAGs can have the same MAG. Similar to the DAGs, a MAG can be identified only up to a family of MAGs that are Markov equivalent. This Markov equivalence class is represented by a *partial ancestral graph* (PAG).

**Markov Equivalence:** Two MAGs are Markov equivalent if and only if they have (i) the same adjacencies; (ii) the same unshielded colliders; and (iii) if a path $\pi$ is a discriminating path for $Z$ in both graphs, then $Z$ is a collider on $\pi$ in one graph if and only if it is a collider on $\pi$ in the other graph as well. A PAG represents the MEC of a MAG that can be learned from the observational data. The skeletons of all MAGs in the MEC are identical. Therefore, the PAG has the same skeleton as all members of the MEC. If an edge is oriented as $\rightarrow$ or $\leftrightarrow$, this orientation is fixed for that edge in all MAGs of the MEC. If an edge in a PAG is oriented as $\leftarrow\!\circ$, this implies that there are at least two MAGs in the MEC, such that for the first MAG, this edge is oriented as $\leftrightarrow$ and for the second MAG, this edge is oriented as $\leftarrow$. An edge with circles on both ends means there are three MAGs in the MEC with three distinct orientations $\leftarrow$, $\rightarrow$, and $\leftrightarrow$.

We denote the MAG corresponding to the DAG $\mathcal{D} = (\mathbf{V} \cup \mathbf{L}, \mathbf{E})$ by $\mathcal{M}_\mathcal{D}$. Let $\mathsf{pa}(A)$, $\mathsf{ch}(A)$, $\mathsf{sp}(A)$, $\mathsf{an}(A)$, and $\mathsf{de}(A)$ denote the sets of parents, children, spouses, ancestors, and descendants of a node $A$. We also create the set $\mathsf{ps}(A) = \mathsf{pa}(A) \cup \mathsf{sp}(A)$ to denote the union of parents and spouses of a node $A$. We denote these relationships with respect to a graph, e.g., $\mathsf{pa}_\mathcal{D}(A)$. The subscript is dropped if the specified graph is clear from the context.

# 4 PROBLEM STATEMENT

Interventions on causal models improve the identifiability of the underlying causal structure. We consider a soft intervention model, which changes the conditional distributions of an intervention target node given its true parents (both observed and unobserved) in the causal DAG $\mathcal{D}$ without completely removing the causal effects of its parents.

**Soft Intervention Model.** Assume that we have $n$ interventional settings, and denote the collection of the intervention target sets by $\mathcal{I} \triangleq \{\mathbf{I}^{(j)} : j \in [n]\}$. In the $j$-th setting, the nodes in $\mathbf{I}^{(j)} \subset \mathbf{V}$ are targeted for intervention. Soft interventions in the linear SEM specified in (1) change the conditional distributions of variables $\{X_i : i \in \mathbf{I}^{(j)}\}$.

Under these changes (i) the variances of the noise terms $\{\epsilon_i : i \in \mathbf{I}^{(j)}\}$ change, and (ii) the weights connecting the parents of the nodes associated with $\{X_i : i \in \mathbf{I}^{(j)}\}$ in the linear SEM *may* change. In other words, $\{B_{\mathsf{pa}(i),i} : i \in \mathbf{I}^{(j)}\}$, where $B_{\mathsf{pa}(i),i} \triangleq \{B_{u,i} : X_u \in \mathsf{pa}(X_i)\}$, may vary freely. We also note that this formulation can readily work with mean-shift interventions that change the mean of the noise variables (see supplementary material Section D.1 for details).

Post-intervention linear SEMs have new parameters. We denote the linear SEM parameters associated with interventions $\mathbf{I}^{(j)}$ by $B^{(j)}$ and $\Omega^{(j)} \triangleq \mathsf{diag}\left((\sigma_1^{(j)})^2, \ldots, (\sigma_p^{(j)})^2\right)$. Since the noise variance terms change under soft interventions, $\mathbf{I}^{(j)}$ is described as follows:

$$\mathbf{I}^{(j)} \triangleq \{i : i \in \mathbf{V}, \ \sigma_i^{(j)} \neq \sigma_i\}. \tag{4}$$

A node can be targeted in multiple interventional settings, e.g., $i \in \mathbf{I}^{(j)} \cap \mathbf{I}^{(l)}$. We assume that each target set have different mechanisms such that upon interventions on sets $\mathbf{I}^{(j)}$ and $\mathbf{I}^{(l)}$, we have $\sigma_i^{(j)} \neq \sigma_i^{(l)}$ for all $i \in \mathbf{I}^{(j)} \cup \mathbf{I}^{(l)}$. We note that this assumption is purely for simplicity in the notation, and can be dropped by denoting the exact mechanism applied on a node under each setting.

Identifiability conditions of the causal graphs with unknown soft interventions and the corresponding graphical characterization are established by Jaber et al. [2020]. Importantly, causal graphs with the same observed variables but different latent variables and intervention targets can still belong to the same MEC. We follow the augmented graph construction of Kocaoglu et al. [2019] and Jaber et al. [2020] to represent the MEC's under interventions graphically. First, we construct the augmented graph as follows: for each pair of intervention targets $\mathbf{I}, \mathbf{J} \in \mathcal{I}$, the augmented graph $\mathsf{Aug}_{\mathcal{I}}(\mathcal{D})$ appends the causal graph $\mathcal{D}$ with an auxiliary node and assign directed edges from this node to each node in $\mathbf{H} = \mathbf{I} \cup \mathbf{J}$. We denote the set of these auxiliary nodes by $\mathcal{F}$, and refer to the members of $\mathcal{F}$ as $F$-nodes. In the example in Fig. 1, we have observational setting $\emptyset$ and intervention target set $\{X\}$, so there is just one pair of target sets. An $F$-node is created corresponding to this pair, and the edge $F \rightarrow X$ is drawn since $X$ is the only node in the set $\emptyset \cup \{X\}$.

**Definition 1 (Augmented Graph)** *Consider a causal graph $\mathcal{D} = (\mathbf{V} \cup \mathbf{L}, \mathbf{E})$ and a set of intervention targets $\mathcal{I}$. Define the multiset $\mathcal{H}$ as $\mathcal{H} = \{\mathbf{I} \cup \mathbf{J} : \mathbf{I}, \mathbf{J} \in \mathcal{I}\}$. Given $\mathcal{H}$, generate $h \triangleq |\mathcal{H}|$ nodes $\mathcal{F} \triangleq \{F_i : i \in [h]\}$ and define the augmented graph of $\mathcal{D}$ as $\mathsf{Aug}_{\mathcal{I}}(\mathcal{D}) \triangleq (\mathbf{V} \cup \mathbf{L} \cup \mathcal{F}, \mathbf{E} \cup \mathcal{E})$, where $\mathcal{E} \triangleq \{(F_i, V) : i \in [h], \ V \in \mathbf{H}_i\}$.*

The study in citetsoftunknown20 shows that the augmented graph exactly represents the separation statements among the random variables in interventional settings. Similar to obtaining a unique MAG from a DAG, a corresponding maximal ancestral graph for the augmented graph is constructed next. In the example in Fig. 1, $F \rightarrow W$ and $Z \rightarrow W$

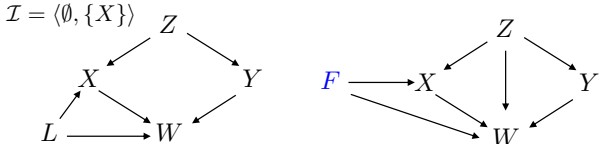

Figure 1: An example of a $\langle \mathcal{D}, \mathcal{I} \rangle$ with $\mathbf{L} = \{L\}$, and the corresponding $\mathcal{I}$-MAG, $\mathcal{M} = \mathsf{MAG}(\mathsf{Aug}_{\mathcal{I}}(\mathcal{D}))$. Note that $F \to X$ is constructed in $\mathsf{Aug}_{\mathcal{I}}(\mathcal{D})$. $F \to W$ edge on $\mathcal{M}$ is due to the inducing path $F \to X \leftarrow L \to W$. Similarly, $Z \to W$ is due to the inducing path $Z \to X \leftarrow L \to W$.

edges are drawn due to the inducing paths existing in the augmented graph $\mathsf{Aug}_{\mathcal{I}}(\mathcal{D})$.

**Definition 2 ($\mathcal{I}$-MAG)** *Given a causal graph $\mathcal{D} = (\mathbf{V} \cup \mathbf{L}, \mathbf{E})$ and a set of intervention targets $\mathcal{I}$, we define $\mathcal{I}$-MAG to represent the maximal ancestral graph constructed over $\mathbf{V}$ from $\mathsf{Aug}_{\mathcal{I}}(\mathcal{D})$, i.e., $\mathsf{MAG}(\mathsf{Aug}_{\mathcal{I}}(\mathcal{D}))$, and denote its edges by $\mathcal{E}_{\mathcal{I}}$.*

Corresponding to every pair of intervention sets $\mathbf{I}^{(j)}$ and $\mathbf{I}^{(l)}$, define the set $\mathbf{I}_{jl} \triangleq \mathbf{I}^{(j)} \cup \mathbf{I}^{(l)}$. Denote the single $F$-node associated with $\mathbf{I}^{(j)}$ and $\mathbf{I}^{(l)}$ by $F_{jl} \in \mathcal{F}$, and denote the set of nodes adjacent to $F_{jl}$ in $\mathcal{I}$-MAG by

$$\mathbf{K}_{jl} \triangleq \{i : (F_{jl}, i) \in \mathcal{E}_{\mathcal{I}}\} . \quad (5)$$

We remark that, in causally sufficient systems, $\mathbf{K}_{jl} = \mathbf{I}_{jl}$. However, in the presence of latent variables, one cannot distinguish between the nodes in $\mathbf{K}_{jl} \setminus \mathbf{I}_{jl}$ and $\mathbf{I}_{jl}$ according to the $\mathcal{I}$-MAG. Therefore, we will focus on estimating $\mathbf{K}_{jl}$, which we call the *effective intervention targets*.

We note that the observational setting can be considered as an interventional setting with an empty target set. When there exist more than two interventional settings, there are multiple $F$-nodes and intervention targets. Accordingly, we denote the set of intervention targets by

$$\mathcal{K} \triangleq \left\{ \mathbf{K}_{jl} : \forall j, l \in [n], \ j \neq l \right\} . \quad (6)$$

**Problem Statement.** We focus on two estimation problems. In the first problem, we estimate the set of intervention targets $\mathcal{K}$ given the data from linear SEMs with latent variables under soft interventions. We denote the estimate of $\mathcal{K}$ by $\hat{\mathcal{K}}$. Our objective is to design the estimator $\phi : \left( \mathbb{R}^{m \times |\mathbf{V}|} \right)^n \to \left( 2^{\mathbf{V}} \right)^n$, in which $|\mathbf{V}|$ denotes the number of the observed variables, $n$ denotes the interventional settings, and $m$ denotes the number of samples in a setting.

In the second problem, based on the estimate $\hat{\mathcal{K}}$, for any set $\mathbf{K} \in \hat{\mathcal{K}}$, we consider the problem of estimating the parents and spouses of $\mathbf{K}$ in the augmented MAG ($\mathcal{I}$-MAG). For any $\mathbf{K} \in \hat{\mathcal{K}}$, we denote the set of parents and spouses of the nodes in $\mathbf{K}$ by $\mathsf{ps}(\mathbf{K})$, and denote its estimate by $\hat{\mathsf{ps}}(\mathbf{K})$.

Therefore, our second objective is to design the estimator $\phi_{\mathsf{ps}(\mathbf{K})} : 2^{\mathbf{V}} \to \left( 2^{\mathbf{V}} \right)^{|\mathbf{K}|}$. These estimates (i.e., $\hat{\mathcal{K}}$ and $\{\hat{\mathsf{ps}}(\mathbf{K}) : \mathbf{K} \in \hat{\mathcal{K}}\}$) are sufficient to refine the observational PAG to the MEC of the $\mathcal{I}$-MAG. In the rest of the paper, we denote this interventional refinement of observational PAG by $\psi$-PAG.

# 5 MAIN RESULTS AND ALGORITHM

**Overview.** In this section, we provide our theoretical results and our **Pre**cision **Di**fference-based Intervention **T**arget **E**stimato**r** (PreDITEr) algorithm. With scalability as the central objective, we focus on estimating only the effective intervention targets. This is a computationally simpler task compared to learning the causal structure of a DAG, and, consequently, facilitates scalability.

The pivotal idea in our algorithm's design is that soft interventions result in only sparse changes in the precision matrix of the linear SEM. Hence, the precision matrix differences have traces of the identities of the intervention sites. We analytically establish how to use the precision matrix differences between a pair of interventional settings to identify the underlying intervened sites. Upon establishing this property, we then devise an algorithm that successively identifies pairs of intervention settings and estimates the difference between their associated precision matrices. These successive estimates are aggregated to identify the intervention targets. Given the extensive literature on estimating precision matrix differences, we can adopt any generic precision difference estimation (PDE) algorithm to generate the estimates that we need in our algorithm.

Once we estimate the target intervention sites, we also provide an estimate for the set of parents and spouses of each of the nodes deemed to be an intervened node. Theoretically, this information enables the increased identifiability of the causal structure due to the interventions. We start describing the details by introducing the precision difference estimation procedure.

**Precision Difference Estimation (PDE).** When the difference between two linear SEMs is sparse, the difference of their respective precision matrices will also be sparse. Hence, for the two intervention target sets $\mathbf{I}^{(j)}$ and $\mathbf{I}^{(l)}$, the difference between their precision matrices $\Delta_{jl} \triangleq \Theta^{(j)} - \Theta^{(l)}$ will be sparse. In this paper, we use the algorithm of Jiang et al. [2018] to estimate $\Delta_{jl}$. The algorithm computes sample covariance matrices $\hat{\Sigma}^{(j)}$ and $\hat{\Sigma}^{(l)}$ from the data. Then, it solves the following convex optimization problem with the alternating direction method of multipliers (ADMM):

$$\hat{\Delta}_{jl} = \operatorname*{argmin}_{\Delta_{jl}} \left\{ \frac{1}{2} \mathsf{Tr}(\Delta_{jl}^{\top} \hat{\Sigma}^{(j)} \Delta_{jl} \hat{\Sigma}^{(l)}) \right.$$
$$\left. - \mathsf{Tr}(\Delta_{jl}(\hat{\Sigma}^{(j)} - \hat{\Sigma}^{(l)})) + \lambda \|\Delta_{jl}\|_1 \right\}, \quad (7)$$

where $\lambda$ is a tuning parameter. We note that there exist alternative approaches to PDE [Zhao et al., 2014, Yuan et al., 2017]. Any method that is guaranteed to converge to the correct solution can be used as our PDE subroutine in a modular way. We have chosen the method of Jiang et al. [2018] due to its significant advantage in computational complexity compared to the others ($O(p^3)$ vs. $O(p^4)$). Next, we define the marginal SEM over a subset of observed variables.

**Definition 3 (Marginal SEM)** *Corresponding to a subset of nodes $S \subseteq \mathbf{V}$, we define $(B_S, \epsilon_S)$ as the marginal SEM that characterizes the relationship among the random variables $X_S \triangleq \{i : i \in S\}$. Accordingly, the corresponding precision matrix is denoted by $\Theta_S$. The parametrization of a marginal SEM is given by the following lemma.*

**Lemma 1 (Ghoshal et al. [2021])** *Corresponding to a subset $S \subseteq \mathbf{W}$, denote the removed set of nodes by $U \triangleq \mathbf{W} \setminus S$ and define $U_i \triangleq U \cap \mathsf{an}(i)$, for $i \in S$. For $i, j \in S$, we have*

$$\sigma_{S,i}^2 = \sigma_i^4 \left( \sigma_i^2 - B_{U_i,i}^\top [\Theta_{\mathsf{an}(i)}]_{U_i,U_i}^{-1} B_{U_i,i} \right)^{-1}, \quad (8)$$

$$[B_S]_{j,i} = \frac{\sigma_{S,i}^2}{\sigma_i^2} \left( B_{j,i} - B_{U_i,i}^\top [\Theta_{\mathsf{an}(i)}]_{U_i,U_i}^{-1} [\Theta_{\mathsf{an}(i)}]_{U_i,j} \right). \quad (9)$$

Before describing the theoretical results, we need the following faithfulness assumption. This assumption rules out the pathological cases in which the effect of an intervention is canceled by other changes in the system. Faithfulness assumptions are generally needed for successful learning.

**Assumption 1 ($\mathcal{I}$-faithfulness)** *For any choice of $i, j \in S \subseteq \mathbf{V}$, we have the following properties:*

- *If $\sigma_i^{(1)} \neq \sigma_i^{(2)}$, then $\sigma_{S,i}^{(1)} \neq \sigma_{S,i}^{(2)}$.*

- *If $\sigma_{S,i}^{(1)} \neq \sigma_{S,i}^{(2)}$, then $[\Theta_S^{(1)}]_{i,i} \neq [\Theta_S^{(2)}]_{i,i}$. If further $[B_S]_{j,i} \neq 0$ in either model, then $[\Theta_S^{(1)}]_{i,j} \neq [\Theta_S^{(2)}]_{i,j}$.*

## 5.1 THEORETICAL RESULTS

For the rest of the discussion, we consider a pair of interventional settings. Without loss of generality, let them be $\mathbf{I}^{(1)}$ and $\mathbf{I}^{(2)}$. Denote the difference in their precision matrices by $\Delta_{12} = \Theta^{(1)} - \Theta^{(2)}$, and the difference in marginal precision matrices for set $S$ by $\Delta_{12_S} = \Theta_S^{(1)} - \Theta_S^{(2)}$. For simplicity in the notation, we denote the corresponding $F$-node $F_{12}$ by $F$, $\mathbf{K}_{12}$ by $\mathbf{K}$, $\Delta_{12}$ by $\Delta$, and $\Delta_{12_S}$ by $\Delta_S$. We also denote the set of *affected nodes* among the observed variables by $S_\Delta \triangleq \{i : [\Delta_{\mathbf{V}}]_{i,i} \neq 0\}$.

**Separation Property for Invariance.** For a non-intervened node $i \in \mathbf{V} \setminus \mathbf{K}$, there is no edge between $F$

and $i$ in $\mathcal{I}$-MAG. Therefore, there exists a set $S$ that separates $F$ and $i$, and the conditional probability distribution of $X_i$ is invariant given $S \setminus \{X_i\}$. Then, the conditional mean and variance of $X_i$, and subsequently $\sigma_{S,i}$, are invariant. Finally, applying the result of Wang et al. [2018], $[\Theta_S]_{i,i} = \sigma_{S,i}^{-2}$ is also invariant. Therefore, the set $S$ that separates $F$ and $i$ yields $[\Delta_S]_{i,i} = 0$ by the definition of $\Delta_S$.

**Theorem 1** *Consider an $F \in \mathcal{F}$ and an observed node $V \in \mathbf{V}$ in the augmented MAG ($\mathcal{I}$-MAG). Then, $(F, V) \in \mathcal{E}_\mathcal{I}$ if and only if $\nexists\, S \subseteq \mathbf{V}$ such that $[\Delta_S]_{V,V} = 0$.*

Theorem 1 states the existence of a conditioning set $S$ for any non-intervened node $V$, that makes the corresponding diagonal entry of the precision matrix invariant. In the following lemma, we show that the ancestors of $V$ within the set of affected nodes $S_\Delta$ suffice to separate $F$ and $V$.

**Lemma 2** *For a node $V \in S_\Delta \setminus \mathbf{K}$, consider the set $S = S_\Delta \cap \mathsf{an}(V)$. Diagonal entry corresponding to $V$ in the precision matrix of the marginal SEM over $S$ is invariant, i.e., $[\Delta_S]_{V,V} = 0$.*

Lemma 2 implies that we can eliminate all the non-intervened nodes (i.e., nodes not in the effective intervention target set $\mathbf{K}$) in $S_\Delta$ by computing PDE for each subset of $S_\Delta$. Therefore, we can identify $\mathbf{K}$ with $2^{|S_\Delta|}$ number of PDEs. Now that we have a way to recover $\mathbf{K}$, we show how to identify the parents and/or spouses of the intervened nodes. This property will play a critical role in improving the identifiability of the MAGs under interventions.

**Lemma 3** *Consider $K \in \mathbf{K}$ and $J \in \mathbf{V} \setminus \mathbf{K}$. If $K \leftarrow\!\circ J$ in $\mathcal{I}$-MAG, there does not exist $S \subseteq S_\Delta$ containing $\{K, J\}$ such that $[\Delta_S]_{K,J} = 0$. On the other hand, if $K \to J$, or there is no edge between them in $\mathcal{I}$-MAG, there exists a set $S \subseteq S_\Delta$ containing $\{K, J\}$ such that $[\Delta_S]_{K,J} = 0$.*

Lemma 2 and Lemma 3 are sufficient to design our algorithm for learning $\mathcal{K}$.

## 5.2 LEARNING ALGORITHM

We leverage the results in Lemma 2 and Lemma 3 to learn the intervention targets $\mathcal{K}$ from a tuple of interventional distributions generated by some unknown pair $\langle \mathcal{D}, \mathcal{I} \rangle$. Algorithm 1 presents our main learning algorithm PreDITEr that uses the results to learn $\mathcal{K}$, and subsequently $\mathsf{ps}(\mathbf{K})$ for $\mathbf{K} \in \mathcal{K}$. We briefly describe PreDITEr and the rationale underlying its design.

Algorithm 1 (PreDITEr) takes sample covariance matrices of interventional data as inputs. Since estimating intervention targets $\mathbf{K}$ for each pair of interventional settings is independent, we investigate each pair individually. For each

**Algorithm 1** Precision Difference-based Intervention Target Estimator (PreDITEr)

1: **Input:** Observed nodes $\mathbf{V}$, sample covariance matrices $\hat{\Sigma}^{(1)}, \ldots, \hat{\Sigma}^{(n)}$
2: **Output.** Intervention targets $\mathcal{K}$, and $\mathsf{ps}(K)$ $\forall K \in \mathbf{K}$, $\forall \mathbf{K} \in \mathcal{K}$
3: $\mathcal{K} \leftarrow \emptyset$, $\mathcal{F} \leftarrow \emptyset$
4: **for** $V \in \mathbf{V}$ **do** $\mathsf{ps}(V) \leftarrow \emptyset$ **end for**
5: **for** all pairs $j, l \in [n]$ **do**
6:      $\mathcal{F} \leftarrow \mathcal{F} \cup \{F_{jl}\}$, $\mathbf{K}_{jl} \leftarrow \emptyset$
7:      Estimate $\Delta_{jl} \leftarrow$ PDE $(\hat{\Sigma}^{(j)}, \hat{\Sigma}^{(l)})$
8:      $S_\Delta \leftarrow \{V : V \in \mathbf{V}, [\Delta_{jl}]_{V,V} \neq 0\}$
9:      For all $S \subseteq S_\Delta$, estimate $\Delta_{jl_S} \leftarrow$ PDE $(\hat{\Sigma}^{(j)}_{S,S}, \hat{\Sigma}^{(l)}_{S,S})$
10:      **for** $V \in \mathbf{V}$ **do**
11:         **if** $\nexists S \subseteq S_\Delta$, such that $V \in S$, and $[\Delta_S]_{V,V} = 0$ **then**
12:            $\mathbf{K}_{jl} \leftarrow \mathbf{K}_{jl} \cup \{V\}$
13:         **end if**
14:      **end for**
15:      $\mathcal{K} \leftarrow \mathcal{K} \cup \mathbf{K}_{jl}$
16:      **for** all pairs $K \in \mathbf{K}_{jl}$, $J \in S_\Delta \setminus \mathbf{K}_{jl}$ **do**
17:         **if** $\nexists S \subseteq S_\Delta$, such that $K, J \in S$, and $[\Delta_S]_{K,J} = 0$ **then**
18:            $\mathsf{ps}(K) \leftarrow \mathsf{ps}(K) \cup \{J\}$
19:         **end if**
20:      **end for**
21: **end for**

---

**Precision Difference Estimation (PDE)** $(\hat{\Sigma}^{(j)}, \hat{\Sigma}^{(l)})$

1: Estimate $\Delta_{jl} = (\hat{\Sigma}^{(j)})^{-1} - (\hat{\Sigma}^{(l)})^{-1}$ using algorithm of Jiang et al. [2018].
2: Symmetrize $\Delta_{jl}$: set $\Delta_{jl} = (\Delta_{jl} + \Delta_{jl}^\top)/2$.
3: Threshold $\Delta_{jl}$: set $[\Delta_{jl}]_{u,v} = 0$ if $|[\Delta_{jl}]_{u,v}| < \varepsilon$.
4: **Return** $\Delta_{jl}$

---

pair of interventional distributions (or the corresponding $F$-node), we first estimate the set of affected nodes $S_\Delta$ (lines 7 and 8). Then, we estimate precision difference $\Delta_S$ for each subset $S$ of $S_\Delta$. If there does not exist a set $S$ for a node $V \in S_\Delta$ such that $[\Delta_S]_{V,V} = 0$, then by Lemma 2, $V$ is an intervened node and belongs to $\mathbf{K}$ (lines 10-15).

After identifying $\mathbf{K}$, consider a $K \in \mathbf{K}$ and $J \in S_\Delta \setminus \mathbf{K}$. If there does not exist a set $S$ such that $[\Delta_S]_{K,J} = 0$, by Lemma 3, $J$ belongs to $\mathsf{ps}(K)$ (lines 16-20).

Algorithm 1 uses PDE as a subroutine. Hence, the quality of the estimate formed by Algorithm 1 hinges on those of the precision difference estimates. To assess the accuracy of Algorithm 1 in estimating the intervention targets irrespectively of the PDE subroutine used, we provide population-level results. In the following theorem, we establish that Algorithm 1 has perfect estimation if the underlying PDE subroutine performs perfectly. This result allows decoupling the accuracy of Algorithm 1 from that of the PDE subroutine

used. In practice, however, PDE subroutines are imperfect, which is imposed by having access to only finite samples. To address the convergence to the correct estimates, we discuss the sample complexity and convergence guarantees of the algorithm of Jiang et al. [2018] in supplementary material Section B.

**Theorem 2** *When the covariance estimates are perfect and Assumption 1 holds, Algorithm 1 perfectly estimates the set of effective intervention targets $\mathcal{K}$ under soft interventions with probability 1. Furthermore, Algorithm 1 recovers non-intervened parents and/or spouses (i.e., $\mathsf{ps}(K)$) of an intervened node $K$ with probability 1.*

## 5.3 RECOVERING $\psi$-MARKOV EQUIVALENCE

Next, we show how we can use the intervention target recovery of Algorithm 1 to refine the observational MEC represented by a PAG to the interventional MEC for soft interventions $\psi$-PAG. We first review the interventional equivalence characterization approaches in the existing literature.

The $\psi$-Markov equivalence property, i.e., the conditions for two $\mathcal{I}$-MAGs to be Markov equivalent, is characterized by Jaber et al. [2020, Theorem 1]. For two MAGs $\mathcal{M}_1$ and $\mathcal{M}_2$ to be $\psi$-Markov equivalent:

- $\mathcal{M}_1$ and $\mathcal{M}_2$ must have the same skeleton.
- $\mathcal{M}_1$ and $\mathcal{M}_2$ must have the same unshielded colliders.
- If a path $\pi$ is a discriminating path for a node $V$ in both $\mathcal{M}_1$ and $\mathcal{M}_2$, then $V$ is a collider on the path in one graph if and only if it is a collider on the path in the other.

The following theorem builds on the results of Theorem 2 and Lemma 3 to obtain $\psi$-PAG.

**Theorem 3 ($\psi$-PAG)** *Given the PAG for the MAG $\mathcal{M}$, and the results of Algorithm 1, i.e., the sets $\mathcal{K}$, $\mathsf{ps}(\mathbf{K})$ $\forall \mathbf{K} \in \mathcal{K}$, we can obtain $\psi$-PAG of $\mathcal{I}$-MAG.*

## 6 EMPIRICAL RESULTS

First, we run our PreDITEr algorithm on synthetically generated data from linear SEMs to recover intervention targets. Next, we provide comparisons with the state-of-the-art method. Finally, we apply our method to a biological dataset to illustrate its applicability to real data. [2]

---

[2]Codebase for reproducing the simulations are available at https://https://github.com/bvarici/uai2022-intervention-estimation-latents

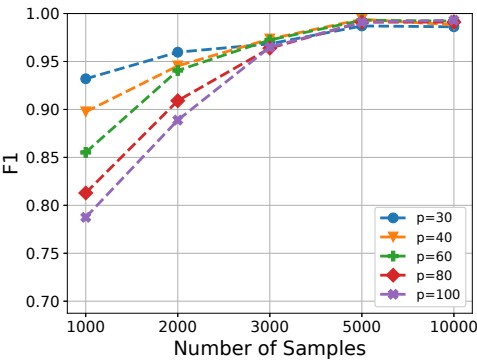

Figure 2: Average F1 scores at estimating **K** for $|\mathbf{L}| = 5$, $|\mathbf{I}| = 5$ intervention targets.

## 6.1 SYNTHETIC DATA

We test the efficiency of PreDITEr for recovering the intervention targets. We generate 100 realizations of Erdős-Rényi random DAGs with the expected neighborhood size $c = 2$. We consider one interventional setting in addition to the observational one, i.e., $\mathcal{I} = \langle \emptyset, \mathbf{I} \rangle$. Therefore, we are estimating a single target set **K**. For each model, we set the number of latent variables to $|\mathbf{L}| = 5$, and the number of intervened nodes to $|\mathbf{I}| = 5$. The edge weights of the causal model, i.e., the entries of $B$, are sampled independently at random according to the uniform distribution on $[-1, -0.25] \cup [0.25, 1]$. The additive Gaussian noise terms have distribution $\mathcal{N}(0, I_p)$. The intervention targets are selected randomly from the observed variables **V**. For the intervened nodes $I \in \mathbf{I}$, upon intervention, the variance of the noise term $\epsilon_I$ changes to 2.

We run PreDITEr with a varying number of samples on graphs with varying sizes $p$. Figure 2 illustrates the target recovery performance. Specifically, it shows that our method recovers the intervention target with high F1 scores. We emphasize that PreDITEr can easily process large graphs (e.g., $p = 100$ nodes), and have less than 1 second average runtime for the simulations shown in Figure 2. This scalability is due to its computational complexity of $O(2^{|S_\Delta|})$. Since the size of $S_\Delta$ is determined only by the number of intervened nodes and their parents/spouses, our method is not directly affected by the graph size $p$.

## 6.2 COMPARISON TO THE RELATED WORK

We compare the scalability and accuracy of PreDITEr to those of two competing methods under various settings: the $\psi$-FCI algorithm of Jaber et al. [2020] and the FCI-JCI123 algorithm of Mooij et al. [2020]. We note that both of these algorithms solve a more general problem than the linear SEMs we are considering. To the best of our knowledge, there is no algorithm specifically designed for linear SEMs, and these are the only two methods that can be applied to

our setting. Therefore, we compare our results to those of these two methods.

Jaber et al. [2020] do not provide simulations for graphs that have more than a few nodes since $\psi$-FCI requires an exponentially growing number of conditional independence and invariance tests. Mooij et al. [2020] report experiments with larger graphs, and we compare our algorithm to their FCI-JCI123 algorithm. We focus on scalability and provide additional experiments on small graphs and MEC refinement results in supplementary material section D.2.

To enable comparisons under soft interventions, we adopt *mechanism changes* of Mooij et al. [2020], in which a constant offset is added to the intervention targets (see page 53, Section 5.2 for details). We note that this is different from our model of soft interventions and results in slight degrading of the performance of our algorithm, but since we are using FCI-JCI123 as our benchmark, we adopt its setting.

We consider two environments and one intervention target for simplicity of the comparisons. We generate 30 Erdős-Rényi random DAGs. The probability of an edge being present in the random graphs is set to $2/p$ where $p$ is the number of observed and latent variables. We report the precision and recall rates of both algorithms along with their runtimes in Table 1. While both methods have similar performance, there is a significant discrepancy in their runtime. More importantly, the runtime of FCI-JCI123 becomes prohibitive very quickly, even with graphs with as few as 40 nodes. In contrast, PreDITEr has a significantly lower runtime even though the considered setting (i.e., mechanism changes) is not the setting for which it is designed. All simulations are run on a computer with i7-4960HQ, 16GB 1600MHz RAM.

Table 1: Intervention recovery results and median runtime.

| Method | $p$ | Precision | Recall | Runtime (s) |
|---|---|---|---|---|
| PreDITEr | 20 | 1.0 | 0.83 | < 1 |
| FCI-JCI123 | 20 | 1.0 | 1.0 | 80.9 |
| PreDITEr | 30 | 1.0 | 0.80 | < 1 |
| FCI-JCI123 | 30 | 1.0 | 0.97 | 318.0 |
| PreDITEr | 40 | 1.0 | 0.87 | < 1 |
| FCI-JCI123 | 40 | 0.96 | 0.96 | 1301.9 |

## 6.3 BIOLOGICAL DATA

We apply the PreDITEr algorithm to a real dataset with data from observational and multiple interventional settings. Since PreDITEr estimates the intervention targets and their corresponding parent-spouse sets for each pair of available settings, we combine the findings from each pair and yield a mixed graph estimate of the associated causal structure.

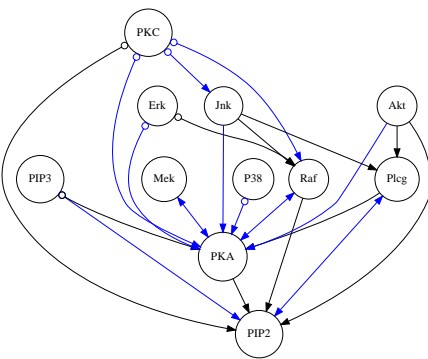

Figure 3: Recovered causal structure using Algorithm 1. Blue edges represent the edges that are in the skeleton of the reference network [Ness et al., 2017].

**Protein signaling data.** We consider the dataset of Sachs et al. [2005], which is a standard benchmark in causal inference literature. The data is obtained from measurements of the proteins involved in T-4 cell signaling. The protein signaling network consists of 11 nodes. In each interventional setting, various drugs are injected into the cells to inhibit or activate different signaling proteins. The target proteins are considered sites of intervention. Data from observational and five interventional settings are provided. The true ground truth network is not exactly known, and the accepted ground truth has been updated over the years. Notably, it is represented by a DAG without latent confounders. We use the recent version of Ness et al. [2017], which consists of 16 edges, and use the preprocessed real data provided by Squires et al. [2020].

Figure 3 shows the output of our algorithm. For a pair of nodes, if they are found to be in the parent-spouse sets of one another, they must be spouses, and we assign a bidirected edge. If only one of them lies in the parent-spouse set of the other node, it must be the parent, and we assign a directed edge to them. If we do not have either of the above results, the relationship can be either parent or spouse, and we denote it by ∘→ on the graph. The recovered edges that are also present in the skeleton of the ground truth DAG are marked in blue. This result illustrates that even though our algorithm is designed for linear models, it has the potential to be applied to real datasets with non-linear models.

## 7 CONCLUSION

In this paper, we have considered the problem of estimating intervention targets for causally insufficient systems in linear structural equation models (SEMs). We have assumed a soft intervention model that is more realistic than hard interventions, which eradicate all causal effects on targets. We have shown the usage of invariance of precision matrix entries and proposed an algorithm to identify intervention targets. The algorithm can also be used to refine the obser-

vational MEC to interventional MEC for maximal ancestral graphs. Since there exist efficient algorithms for the former, our algorithm provides scalability for the latter as well. We support our analytical results through simulations and compare them with competing methods. The limitation of our approach is that it only applies to linear SEMs. However, we have demonstrated strong performance in real and synthetic datasets, which shows its applicability to other settings.

## Acknowledgements

This work was supported by the Rensselaer-IBM AI Research Collaboration (http://airc.rpi.edu), part of the IBM AI Horizons Network (http://ibm.biz/AIHorizons).

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
