# OpenReview forum: "Intervention Target Estimation in the Presence of Latent Variables"
_auai.org/UAI/2022/Conference — UAI 2022 Poster_

### Official Review · Reviewer_T6fy · 2022-03-28

**Q2(1) Originality/Novelty:** 2
**Q2(2) Significance/Impact:** 2
**Q2(3) Correctness/Technical Quality:** 3
**Q2(6) Clarity Of Writing:** 3
**Q6 Overall Score:** 6
**Q8 Confidence In Your Score:** 3

**Q1 Summary And Contributions:**

The authors propose an algorithm (PreDITEr) for finding effective intervention targets given sparse soft interventions in the presence of latent variables. The authors assume linear SEMs with a Gaussian error distribution and use (sparse) changes in the precision makes as basis for their inference. The main contribution is scalability: While sacrificing extensive information on causal structure, the authors focus on the intervention targets only. This makes a dramatic speed-up possible.

**Q2 Assessment Of The Paper:**

More detailed information regarding each of these aspects is given below:

**Q2(4) Quality Of Experiments (Optional):**

2: Fair: The experimental evaluation is weak: important baselines are missing, or the results do not adequately support the main claims.

**Q2(5) Reproducibility:**

3: Good: Key resources (e.g., proofs, code, data) are available and key details (e.g., proofs, experimental setup) are sufficiently well-described for competent researchers to confidently reproduce the main results.

**Q3 Main Strengths:**

- A substantial speed-up was achieved, which makes analysis of problems possible that were until now infeasible.
- The paper is well structured, however, some details could be improved.
- There is an extensive appendix with more details or further information.

**Q4 Main Weakness:**

- In chapter 4, the introduction of augmented graphs and I-MAGs is poorely motivated and thus hard to follow. Moreover, while Fig. 1 presents an example on this topic, this is nowhere exploited in the text. I suggest to guide the reader through the example of Fig. 1 in the text and explain notation this way.
- The concept of \Psi-graph is used on p.5 but is only introduced later. Ideally introduce before using it or at least refer to later definition.
- Similarly the subchapter on PDE in Ch. 5 is not well motivated and especially eq. (7) does not seem to be very informative unless more context is given.
- Simulation wrt. Fig. 2: Please give more infos on the runtime, since scalability is the main focus on this paper.
- Was the PDE subroutine by Jiang et.al. (2018) used in the simulation ? Are there competitors and how would the performance change when using competitors ?
- Can you give the exact sample complexity of your algorithm (as for PDE alone in Appendix B)?

**Q5 Detailed Comments To The Authors:**

- the overall quality of wrting is high, I did not find substantial typos
- My main concern is readability wrt. concepts and inclusion of examples which was detailed in the sections above
- Apart from mere scalability wrt. intervention targets, the proposed algorithm can estimate the set of parents and spouses of the nodes deemed to be an intervened node. From my perspective, this advantage of the algorithm is not exploited in the current text. Perhaps it goes beyond the scope of this paper to explain the potential impact this might have. Still, since this is another selling point of the new algorithm, I would encourage the authors to outline the potential use in causal structure identification and make clear in which settings one could gain an advantage over existing methods for causal structure identification. A simulation study might drive this point home, but as I said, that is probably beyond the scope of this paper.


**Q7 Justification For Your Score:**

- Clear improvement in scalability but at the cost of less information in the output compared to related methods
- No substantial methodological improvements
- Readibility could be improved

**Q9 Complying With Reviewing Instructions:**

1: Yes.

---

### Official Review · Reviewer_pjf7 · 2022-04-08

**Q2(1) Originality/Novelty:** 3
**Q2(2) Significance/Impact:** 2
**Q2(3) Correctness/Technical Quality:** 3
**Q2(6) Clarity Of Writing:** 2
**Q6 Overall Score:** 5
**Q8 Confidence In Your Score:** 2

**Q1 Summary And Contributions:**

The authors propose a method for estimating the targets of certain sof-interventions even when latent confounders exist. The method is experimentally compared with existing methods for a similar purpose.

**Q2 Assessment Of The Paper:**

More detailed information regarding each of these aspects is given below:

**Q2(4) Quality Of Experiments (Optional):**

2: Fair: The experimental evaluation is weak: important baselines are missing, or the results do not adequately support the main claims.

**Q2(5) Reproducibility:**

2: Fair: Key resources (e.g., proofs, code, data) are unavailable but key details (e.g., proof sketches, experimental setup) are sufficiently well-described for an expert to confidently reproduce the main results.

**Q3 Main Strengths:**

The authors present an algorithm for target intervention identification in linear SEMs with latent variables. The soft interventions considered make this possible as they produce a specific pattern in the covariance/precision matrix. The paper is rather theoretical and I did not check the correctness of the claims. I have no reason to believe that they are flawed.

**Q4 Main Weakness:**

The method proposed only works for linear SEMs and a very particular type of soft-interventions. The latter is my main criticism. The authors make no effort to convince the reader that this type of soft-interventions makes sense in reality.

I find misleading the comparison that the authors make between their method and other existing methods. They claim that their method is scalable because it identifies the targets without identifying the whole causal structure, as other existing methods do, e.g. Mooij et al. and Jaber et al. Well, this comparison is a bit unfair because the method proposed in this paper only works for linear SEMs and a very particular type of soft-interventions, whereas the other methods work for general causal models and soft-interventions. So, I think the authors are overselling their method in the introduction and, more importantly, the experimental evaluation in Section 6 is unfair.

**Q5 Detailed Comments To The Authors:**

The method proposed only works for linear SEMs and a very particular type of soft-interventions. The latter is my main criticism. The authors make no effort to convince the reader that this type of soft-interventions makes sense in reality. The authors spend quite some time in Section 1 telling how important target identification is in practice but, then, they assume a type of soft-interventions that imho is quite unrealistic. At least the authors do not say the opposite. This needs to be clarified in the revised version of the manuscript. For instance, the authors say that hard interventions can be difficult to perform in practice. And what about their soft-interventions? Are they easy to perform? I doubt it.

The authors also state in Section 1 that "there is a recent growing interest in using causal discovery for fault localization in systems of microservices in cloud-native applications". They go on to say that "faults are modeled as soft interventions". I checked the two references provided and I could not verify this claim. Sorry if I missed it. In any case, do those soft interventions resemble the ones used in the present manuscript?

I find misleading the comparison that the authors make between their method and other existing methods. They claim that their method is scalable because it identifies the targets without identifying the whole causal structure, as other existing methods do, e.g. Mooij et al. and Jaber et al. Well, this comparison is a bit unfair because the method proposed in this paper only works for linear SEMs and a very particular type of soft-interventions, whereas the other methods work for general causal models and soft-interventions. So, I think the authors are overselling their method in the introduction and, more importantly, the experimental evaluation in Section 6 is unfair.

I do not quite get the paragraph under Equation 4, specifically the sentence about "We note that this assumption ...". Can you please ellaborate further?

The authors state right before Section 7 that "This result illustrates that even though our algorithm is designed for linear models, it can be applied to real datasets with non-linear models, and recovers most of the skeleton correctly." Imho, the authors are not entitled to state that based on just one dataset. This is again overselling their method.

**Q7 Justification For Your Score:**

I appreciate the new theoretical results. However, I find the motivation being somewhat misleading. Imho the soft-interventions considered in the paper are rather unrealistic.

**Q9 Complying With Reviewing Instructions:**

1: Yes.

---

### Official Review · Reviewer_dBtX · 2022-04-12

**Q2(1) Originality/Novelty:** 2
**Q2(2) Significance/Impact:** 2
**Q2(3) Correctness/Technical Quality:** 2
**Q2(6) Clarity Of Writing:** 2
**Q6 Overall Score:** 5
**Q8 Confidence In Your Score:** 3

**Q1 Summary And Contributions:**

In this paper, the authors considered the problem of estimating intervention targets using data from multiple environments. Using the empirical covariance matrices from different environment as sufficient statistics, the authors proposed a new approach to estimate the intervention targets when the data has latent variables.

**Q2 Assessment Of The Paper:**

More detailed information regarding each of these aspects is given below:

**Q2(4) Quality Of Experiments (Optional):**

2: Fair: The experimental evaluation is weak: important baselines are missing, or the results do not adequately support the main claims.

**Q2(5) Reproducibility:**

2: Fair: Key resources (e.g., proofs, code, data) are unavailable but key details (e.g., proof sketches, experimental setup) are sufficiently well-described for an expert to confidently reproduce the main results.

**Q3 Main Strengths:**

1. In Ghoshal et al. and Wang et al., the authors mainly focused on unveiling the underlying causal mechanism with environment shifts, which results in very time consuming algorithms that rely on multiple hypothesis testing for causal discovery. Moreover, the results rely heavily on the no latent variable assumption. In this paper, the authors focus on a less ambitious goal of estimating the intervention targets, thereby allowing the method to be more robust to the existence of latent variables. This is important for example in the discovery of biological networks, where the causal graphs are usually hard to estimate due to the existence of latent confounders.

**Q4 Main Weakness:**

1. However, in Steps 8-11 of algorithm 1, the algorithm still needs to enumerate all the subsets of nodes for intervention target estimation. Hence, I am not very convinced that it can be more efficient than for e.g. the methods proposed by Ghoshal et al. and Wang et al.. More discussions are still needed in order to better compare the time complexity of this approach compared with the previous two methods.

2. Personally, it seems to me the theory and methodology looks a bit similar to the one by Ghoshal et al. and also Wang et al., which may impedes its degree of originality.

**Q5 Detailed Comments To The Authors:**

1. Make more discussions about the time complexity, especially comparisons with Ghoshal et al. and Wang et al.

2. Is it possible to develop some nonparametric version of this method?

**Q7 Justification For Your Score:**

There are some interesting improvements, but the methodology seems quite close to the previous approaches, therefore I vote for borderline accept.

**Q9 Complying With Reviewing Instructions:**

1: Yes.

---

### Official Review · Reviewer_MBhr · 2022-04-18

**Q2(1) Originality/Novelty:** 3
**Q2(2) Significance/Impact:** 2
**Q2(3) Correctness/Technical Quality:** 3
**Q2(6) Clarity Of Writing:** 3
**Q6 Overall Score:** 6
**Q8 Confidence In Your Score:** 2

**Q1 Summary And Contributions:**

Given samples of variables where some variables have been intervened on, they present an algorithm to discover these variables and adjacent edges. From this and a PAG, they can also learn the Markov equivalent class of interventional MAGs.

**Q2 Assessment Of The Paper:**

More detailed information regarding each of these aspects is given below:

**Q2(4) Quality Of Experiments (Optional):**

3: Good: The experimental evaluation is adequate, and the results convincingly support the main claims.

**Q2(5) Reproducibility:**

2: Fair: Key resources (e.g., proofs, code, data) are unavailable but key details (e.g., proof sketches, experimental setup) are sufficiently well-described for an expert to confidently reproduce the main results.

**Q3 Main Strengths:**


A novel method of learning intervention targets

They give proof and perform experiments

**Q4 Main Weakness:**


I am not sure intervention targets are an important problem to solve

A little dry to read with pages of definitions


**Q5 Detailed Comments To The Authors:**


Is it really useful to learn intervention targets? The person doing the interventions would know the targets already



>p3 B ∈ R p×p is the edge weights matrix

Is it not more usual to call the matrix D ?

>p6 Algorithm 6

V is also in the input

> line 6: F ← F ∪ {F_jl },

Why is that F updated and then never used?


>p8 and yield a mixed graph estimate of the associated causal structure.

Is this the entire causal structure? Edges around nodes that are not neighbous of intervened nodes are not discovered by the algorithm, are they?

>of the consensus network.

What is the consensus network?

Is it really useful to learn intervention targets? The person doing the interventions would know the targets already



>p3 B ∈ R p×p is the edge weights matrix

Is it not more usual to call the matrix D ?

>p6 Algorithm 6

V is also in the input

> line 6: F ← F ∪ {F_jl },

Why is that F updated and then never used?


>p8 and yield a mixed graph estimate of the associated causal structure.

Is this the entire causal structure? Edges around nodes that are not neighbous of intervened nodes are not discovered by the algorithm, are they?

>of the consensus network.

What is the consensus network?





Is there code for the experiments?



**Q7 Justification For Your Score:**

See Q3 to Q5

**Q9 Complying With Reviewing Instructions:**

1: Yes.

---

### Decision · Program_Chairs · 2022-05-15

**Decision:**

Accept (Poster)

**Comment:**

Meta Review: This paper proposes a method to estimate the intervention targets using data from multiple regimes featuring soft interventions with unknown intervention targets, assuming linear models with possibly latent variables, and a way to infer the non-intervened parents or spouses of an intervened variable. The method is based on searching for certain kinds of invariance across settings, by estimating differences in the precision matrices. The central idea is not new, but a new algorithm is devised and shown to be theoretically sound and empirically more scalable than some previous methods.

All reviewers are more or less inclined to judge this paper acceptable: two "weak accept" and two "borderline accept". A concern raised by some reviewers is that the novelty of the paper is in fact quite limited. I share this judgment, but I think the new algorithm is still a useful and worthy contribution, given the importance of accommodating the possibility of latent confounders in such multi-regime settings. The paper is overall readable, but as some reviewers pointed out, some parts, e.g., the description of the PDE algorithm used in this paper and that of interventional MEC in Section 5.3, should be made clearer.

I also suggest the authors make more explicit in Section 2 what they acknowledge in their response to reviewers, that this paper is essentially another attempt to infer causal information by an invariance criterion. Note that the idea is much order than Peters et al. (2016) and can be traced back to Hoover (1990, "The logic of causal inference") and Jin and Judea (2001, "Causal discovery from changes"). The invariance approach was also generalized to independence-of-change or minimal-change approach in e.g., Ghassami et al. (2018, Multi-domain causal structure learning in linear systems) and Huang et al. (2020, Causal discovery from heterogeneous/nonstationary data), which should be noted in the literature review.